# A Novel Two-Step Method for Cross Language Representation Learning

**Min Xiao** and **Yuhong Guo**
Department of Computer and Information Sciences
Temple University, Philadelphia, PA 19122, USA
{minxiao,yuhong}@temple.edu

## Abstract

Cross language text classification is an important learning task in natural language processing. A critical challenge of cross language learning arises from the fact that words of different languages are in disjoint feature spaces. In this paper, we propose a two-step representation learning method to bridge the feature spaces of different languages by exploiting a set of parallel bilingual documents. Specifically, we first formulate a matrix completion problem to produce a complete parallel document-term matrix for all documents in two languages, and then induce a low dimensional cross-lingual document representation by applying latent semantic indexing on the obtained matrix. We use a projected gradient descent algorithm to solve the formulated matrix completion problem with convergence guarantees. The proposed method is evaluated by conducting a set of experiments with cross language sentiment classification tasks on Amazon product reviews. The experimental results demonstrate that the proposed learning method outperforms a number of other cross language representation learning methods, especially when the number of parallel bilingual documents is small.

## 1 Introduction

Cross language text classification is an important natural language processing task that exploits a large amount of labeled documents in an auxiliary source language to train a classification model for classifying documents in a target language where labeled data is scarce. An effective cross language learning system can greatly reduce the manual annotation effort in the target language for learning good classification models. Previous work in the literature has demonstrated successful performance of cross language learning systems on various cross language text classification problems, including multilingual document categorization [2], cross language fine-grained genre classification [14], and cross-lingual sentiment classification [18, 16].

The challenge of cross language text classification lies in the language barrier. That is documents in different languages are expressed with different word vocabularies and thus have disjoint feature spaces. A variety of methods have been proposed in the literature to address cross language text classification by bridging the cross language gap, including transforming the training or test data from one language domain into the other language domain by using machine translation tools or bilingual lexicons [18, 6, 23], and constructing cross-lingual representations by using readily available auxiliary resources such as bilingual word pairs [16], comparable corpora [10, 20, 15], and other multilingual resources [3, 14].

In this paper, we propose a two-step learning method to induce cross-lingual feature representations for cross language text classification by exploiting a set of unlabeled parallel bilingual documents. First we construct a concatenated bilingual document-term matrix where each document is represented in the concatenated vocabulary of two languages. In such a matrix, a pair of parallel

documents are represented as a row vector filled with observed word features from both the source language domain and the target language domain, while a non-parallel document in a single language is represented as a row vector filled with observed word features only from its own language and has missing values for the word features from the other language. We then learn the unobserved feature entries of this sparse matrix by formulating a matrix completion problem and solving it using a projected gradient descent optimization algorithm. By doing so, we expect to automatically capture important and robust low-rank information based on the word co-occurrence patterns expressed both within each language and across languages. Next we perform latent semantic indexing over the recovered document-term matrix and induce a low-dimensional dense cross-lingual representation of the documents, on which standard monolingual classifiers can be applied. To evaluate the effectiveness of the proposed learning method, we conduct a set of experiments with cross language sentiment classification tasks on multilingual Amazon product reviews. The empirical results show that the proposed method significantly outperforms a number of cross language learning methods. Moreover, the proposed method produces good performance even with a very small number of unlabeled parallel bilingual documents.

## 2   Related Work

Many works in the literature address cross language text classification by first translating documents from one language domain into the other one via machine translation tools or bilingual lexicons and then applying standard monolingual classification algorithms [18, 23], domain adaptation techniques [17, 9, 21], or multi-view learning methods [22, 2, 1, 13, 12]. For example, [17] proposed an expectation-maximization based self-training method, which first initializes a monolingual classifier in the target language with the translated labeled documents from the source language and then retrains the model by adding unlabeled documents from the target language with automatically predicted labels. [21] proposed an instance and feature bi-weighting method by first translating documents from one language domain to the other one and then simultaneously re-weighting instances and features to address the distribution difference across domains. [22] proposed to use the co-training method for cross language sentiment classification on parallel corpora. [2] proposed a multi-view majority voting method to categorize documents in multiple views produced from machine translation tools. [1] proposed a multi-view co-classification method for multilingual document categorization, which minimizes both the training loss for each view and the prediction disagreement between different language views. Our proposed approach in this paper shares similarity with these approaches in exploiting parallel data produced by machine translation tools. But our approach only requires a small set of unlabeled parallel documents, while these approaches require at least translating all the training documents in one language domain.

Another important group of cross language text classification methods in the literature construct cross-lingual representations by exploiting bilingual word pairs [16, 7], parallel corpora [10, 20, 15, 19, 8], and other resources [3, 14]. [16] proposed a cross-language structural correspondence learning method to induce language-independent features by using pivot word pairs produced by word translation oracles. [10] proposed a cross-language latent semantic indexing (CL-LSI) method to induce cross-lingual representations by performing LSI over a dual-language document-term matrix, where each dual-language document contains its original words and the corresponding translation text. [20] proposed a cross-lingual kernel canonical correlation analysis (CL-KCCA) method. It first learns two projections (one for each language) by conducting kernel canonical correlation analysis over a paired bilingual corpus and then uses them to project documents from language-specific feature spaces to the shared multilingual semantic feature space. [15] employed cross-lingual oriented principal component analysis (CL-OPCA) over concatenated parallel documents to learn a multilingual projection by simultaneously minimizing the projected distance between parallel documents and maximizing the projected covariance of documents across languages. Some other work uses multilingual topic models such as the coupled probabilistic latent semantic analysis and the bilingual latent Dirichlet allocation to extract latent cross-lingual topics as interlingual representations [19]. [14] proposed to use language-specific part-of-speech (POS) taggers to tag each word and then map those language-specific POS tags to twelve universal POS tags as interlingual features for cross language fine-grained genre classification. Similar to the multilingual semantic representation learning approaches such as CL-LSI, CL-KCCA and CL-OPCA, our two-step learning method exploits parallel documents. But different from these methods which apply operations such as LSI, KCCA, and OPCA directly on the original concatenated document-

term matrix, our method first fills the missing entries of the document-term matrix using matrix completion, and then performs LSI over the recovered low-rank matrix.

## 3 Approach

In this section, we present the proposed two-step learning method for learning cross-lingual document representations. We assume a subset of unlabeled parallel documents from the two languages are given, which can be used to capture the co-occurrence of terms across languages and build connections between the vocabulary sets of the two languages. We first construct a unified document-term matrix for all documents from the auxiliary source language domain and the target language domain, whose columns correspond to the word features from the unified vocabulary set of the two languages. In this matrix, each pair of parallel documents is represented as a fully observed row vector, and each non-parallel document is represented as a partially observed row vector where only entries corresponding to words in its own language vocabulary are observed. Instead of learning a low-dimensional cross-lingual document representation from this matrix directly, we perform a two-step learning procedure: First we learn a low-rank document-term matrix by automatically filling the missing entries via matrix completion. Next we produce cross-lingual representations by applying the latent semantic indexing method over the learned matrix.

Let $M^0 \in \mathbb{R}^{t \times d}$ be the unified document-term matrix, which is partially filled with observed nonnegative feature values, where $t$ is the number of documents and $d$ is the size of the unified vocabulary. We use $\Omega$ to denote the index set of the observed features in $M^0$, such that $(i,j) \in \Omega$ if only if $M^0_{ij}$ is observed; and use $\widehat{\Omega}$ to denote the index set of the missing features in $M^0$, such that $(i,j) \in \widehat{\Omega}$ if only if $M^0_{ij}$ is unobserved. For the $i$-th document in the data set from one language, if the document does not have a parallel translation in the other language, then all the features in row $M^0_{i:}$ corresponding to the words in the vocabulary of the other language are viewed as missing features.

### 3.1 Matrix Completion

Note that the document-term matrix $M^0$ has a large fraction of missing features and the only bridge between the vocabulary sets of the two languages is the small set of parallel bilingual documents. Learning from this partially observed matrix directly by treating missing features as zeros certainly will lose a lot of information. On the other hand, a fully observed document-term matrix is naturally low-rank and sparse, as the vocabulary set is typically very large and each document only contains a small fraction of the words in the vocabulary. Thus we propose to automatically fill the missing entries of $M^0$ based on the feature co-occurrence information expressed in the observed data, by conducting matrix completion to recover a low-rank and sparse matrix. Specifically, we formulate the matrix completion as the following optimization problem

$$\min_M \ rank(M) + \mu \|M\|_1 \quad \text{subject to} \ M_{ij} = M^0_{ij}, \forall (i,j) \in \Omega; \ M_{ij} \geq 0, \forall (i,j) \in \widehat{\Omega} \quad (1)$$

where $\| \cdot \|_1$ denotes a $\ell_1$ norm and is used to enforce sparsity. The rank function however is non-convex and difficult to optimize. We can relax it to its convex envelope, a convex trace norm $\|M\|_*$. Moreover, instead of using the equality constraints in (1), we propose to minimize a regularization loss function, $c(M_{ij}, M^0_{ij})$, to cope with observation noise for all the observed feature entries. Meanwhile, we also add regularization terms over the missing features, $c(M_{ij}, 0), \forall (i,j) \in \widehat{\Omega}$, to avoid overfitting. In particular, we use the least squared loss function $c(x,y) = \frac{1}{2} \|x - y\|^2$. Hence we obtain the following relaxed convex optimization problem for matrix completion

$$\min_M \ \gamma \|M\|_* + \mu \|M\|_1 + \sum_{(i,j) \in \Omega} c(M_{ij}, M^0_{ij}) + \rho \sum_{(i,j) \in \widehat{\Omega}} c(M_{ij}, 0) \quad \text{subject to} \ M \geq 0 \quad (2)$$

With nonnegativity constraints $M \geq 0$, the non-smooth $\ell_1$ norm regularizer in the objective function of (2) is equivalent to a smooth linear function $\|M\|_1 = \sum_{ij} M_{ij}$. Nevertheless, with the non-smooth trace norm $\|M\|_*$, the optimization problem (2) remains to be convex but non-smooth. Moreover, the matrix $M$ in cross-language learning tasks is typically very large, and thus a scalable optimization algorithm needs to be developed to conduct efficient optimization. In next section, we will present a scalable projected gradient descent algorithm to solve this minimization problem.

---

**Algorithm 1** Projected Gradient Descent Algorithm

---

**Input:** $M^0$, $\gamma$, $\rho \leq 1$, $0 < \tau < \min(2, \frac{2}{\rho})$, $\mu$.
Initialize $M$ as the nonnegative projection of the rank-1 approximation of $M^0$.
**while** not converged **do**
      1. gradient descent: $M = M - \tau \nabla g(M)$.
      2. shrink: $M = \mathcal{S}_{\tau\gamma}(M)$.
      3. project onto feasible set: $M = \max(M, 0)$.
**end while**

---

## 3.2 Latent Semantic Indexing

After solving (2) for an optimal low-rank solution $M^*$, we can use each row of the sparse matrix $M^*$ as a vector representation for each document in the concatenated vocabulary space of the two languages. However exploiting such a matrix representation directly for cross language text classification lacks sufficient capacity of handling feature noise and sparseness, as each document is represented using a very small set of words in the vocabulary set. We thus propose to apply a latent semantic indexing (LSI) method on $M^*$ to produce a low-dimensional semantic representation of the data. LSI uses singular value decomposition to discover the important associative relationships of word features [10], and create a reduced-dimension feature space. Specifically, we first perform singular value decomposition over $M^*$, $M^* = USV^\top$, and then obtain a low dimensional representation matrix $Z$ via a projection $Z = M^*V_k$, where $V_k$ contains the top $k$ right singular vectors of $M^*$. Cross-language text classification can then be conducted over $Z$ using monolingual classifiers.

# 4 Optimization Algorithm

## 4.1 Projected Gradient Descent Algorithm

A number of algorithms have been developed to solve matrix completion problems in the literature [4, 11]. We use a projected gradient descent algorithm to solve the non-smooth convex optimization problem in (2). This algorithm takes the objective function $f(M)$ in (2) as a composition of a non-smooth term and a convex smooth term such as $f(M) = \gamma\|M\|_* + g(M)$ where

$$g(M) = \mu\|M\|_1 + \sum_{(i,j)\in\Omega} c(M_{ij}, M_{ij}^0) + \rho \sum_{(i,j)\in\widehat{\Omega}} c(M_{ij}, 0). \tag{3}$$

It first initializes $M$ as the nonnegative projection of the rank-1 approximation of $M^0$, and then iteratively updates $M$ using a projected gradient descent procedure. In each iteration, we perform three steps to update $M$. First, we take a gradient descent step $M = M - \tau\nabla g(M)$ with stepsize $\tau$ and gradient function

$$\nabla g(M) = \mu E + (M - M^0) \circ Y + \rho M \circ \widehat{Y} \tag{4}$$

where $E$ is a $t \times d$ matrix with all 1s; $Y$ and $\widehat{Y}$ are $t \times d$ indicator matrices such that $Y_{ij} = 1$ if and only if $(i, j) \in \Omega$ and $\widehat{Y} = E - Y$; and "$\circ$" denotes the Hadamard product. Next we perform a shrinkage operation $M = \mathcal{S}_\nu(M)$ over the resulting matrix from the first step to minimize its rank. The shrinkage operator is based on singular value decomposition

$$\mathcal{S}_\nu(M) = U\Sigma_{(\nu)}V^\top, \quad M = U\Sigma V^\top, \quad \Sigma_{(\nu)} = \max(\Sigma - \nu, 0), \tag{5}$$

where $\nu = \tau\gamma$. Finally we project the resulting matrix into the nonnegative feasible set by $M = \max(M, 0)$. This update procedure provably converges to an optimal solution. The overall algorithm is given in Algorithm 1.

## 4.2 Convergence Analysis

Let $h(\cdot) = I(\cdot) - \tau\nabla g(\cdot)$ be the gradient descent operator used in the gradient descent step, and let $P_\mathcal{C}(\cdot) = \max(\cdot, 0)$ be the projection operator, while $\mathcal{S}_\nu(\cdot)$ is the shrinkage operator. Below we prove the convergence of the projected gradient descent algorithm.

**Lemma 1.** *Let $E$ be a $t \times d$ matrix with all 1s, and $Q = E - \tau(Y + \rho \widehat{Y})$. For $\tau \in (0, \min(2, \frac{2}{\rho}))$, the operator $h(\cdot)$ is non-expansive, i.e., for any $M$ and $M' \in \mathbb{R}^{t \times d}$, $\|h(M) - h(M')\|_F \leq \|M - M'\|_F$. Moreover, $\|h(M) - h(M')\|_F = \|M - M'\|_F$ if and only if $h(M) - h(M') = M - M'$.*

*Proof.* Note that for $\tau \in (0, \min(2, \frac{2}{\rho}))$, we have $-1 < Q_{ij} < 1, \forall(i,j)$. Then following the gradient definition in (4), we have

$$\|h(M) - h(M')\|_F = \|(M - M') \circ Q\|_F = \left(\sum_{ij}(M_{ij} - M'_{ij})^2 Q_{ij}^2\right)^{\frac{1}{2}} \leq \|M - M'\|_F$$

The inequalities become equalities if only if $h(M) - h(M') = M - M'$. $\qquad\square$

**Lemma 2.** *[11, Lemma 1] The shrinkage operator $\mathcal{S}_\nu(\cdot)$ is non-expansive, i.e., for any $M$ and $M' \in \mathbb{R}^{t \times d}$, $\|\mathcal{S}_\nu(M) - \mathcal{S}_\nu(M')\|_F \leq \|M - M'\|_F$. Moreover, $\|\mathcal{S}_\nu(M) - \mathcal{S}_\nu(M')\|_F = \|M - M'\|_F$ if and only if $\mathcal{S}_\nu(M) - \mathcal{S}_\nu(M') = M - M'$.*

**Lemma 3.** *The projection operator $P_\mathcal{C}(\cdot)$ is non-expansive, i.e., $\|P_\mathcal{C}(M) - P_\mathcal{C}(M')\|_F \leq \|M - M'\|_F$. Moreover, $\|P_\mathcal{C}(M) - P_\mathcal{C}(M')\|_F = \|M - M'\|_F$ if and only if $P_\mathcal{C}(M) - P_\mathcal{C}(M') = M - M'$.*

*Proof.* For any given entry index $(i, j)$, there are four cases:

- Case 1: $M_{ij} \geq 0, M'_{ij} \geq 0$. We have $(P_\mathcal{C}(M_{ij}) - P_\mathcal{C}(M'_{ij}))^2 = (M_{ij} - M'_{ij})^2$.

- Case 2: $M_{ij} \geq 0, M'_{ij} < 0$. We have $(P_\mathcal{C}(M_{ij}) - P_\mathcal{C}(M'_{ij}))^2 = M_{ij}^2 < (M_{ij} - M'_{ij})^2$.

- Case 3: $M_{ij} < 0, M'_{ij} \geq 0$. We have $(P_\mathcal{C}(M_{ij}) - P_\mathcal{C}(M'_{ij}))^2 = M'^2_{ij} < (M_{ij} - M'_{ij})^2$.

- Case 4: $M_{ij} < 0, M'_{ij} < 0$. We have $(P_\mathcal{C}(M_{ij}) - P_\mathcal{C}(M'_{ij}))^2 = 0 \leq (M_{ij} - M'_{ij})^2$.

Therefore,

$$\|P_\mathcal{C}(M) - P_\mathcal{C}(M')\|_F = \left(\sum_{ij}(P_\mathcal{C}(M_{ij}) - P_\mathcal{C}(M'_{ij}))^2\right)^{\frac{1}{2}} \leq \left(\sum_{ij}(M_{ij} - M'_{ij})^2\right)^{\frac{1}{2}} = \|M - M'\|_F$$

and $\|P_\mathcal{C}(M) - P_\mathcal{C}(M')\|_F = \|M - M'\|_F$ if only if $P_\mathcal{C}(M) - P_\mathcal{C}(M') = M - M'$. $\qquad\square$

**Theorem 1.** *The sequence $\{M^k\}$ generated by the projected gradient descent iterations in Algorithm 1 with $0 < \tau < \min(2, \frac{2}{\rho})$ converges to $M^*$, which is an optimal solution of (2).*

*Proof.* Since $h(\cdot)$, $\mathcal{S}_\nu(\cdot)$ and $P_\mathcal{C}(\cdot)$ are all non-expansive, the composite operator $P_\mathcal{C}(\mathcal{S}_\nu(h(\cdot)))$ is non-expansive as well. This theorem can then be proved following [11, Theorem 4]. $\qquad\square$

## 5 Experiments

In this section, we evaluate the proposed two-step learning method by conducting extensive cross language sentiment classification experiments on multilingual Amazon product reviews.

### 5.1 Experimental Setting

**Dataset** We used the multilingual Amazon product reviews dataset [16], which contains three categories (Books (B), DVD (D), Music (M)) of product reviews in four different languages (English (E), French (F), German (G), Japanese (J)). For each category of the product reviews, there are 2000 positive and 2000 negative English reviews, and 1000 positive and 1000 negative reviews for each of the other three languages. In addition, there are another 2000 unlabeled parallel reviews between English and each of the other three languages. Each review is preprocessed into a unigram bag-of-word feature vector with TF-IDF values. We focused on cross-lingual learning between English and the other three languages and constructed 18 cross language sentiment classification tasks (EFB, FEB, EFD, FED, EFM, FEM, EGB, GEB, EGD, GED, EGM, GEM, EJB, JEB, EJD, JED, EJM, JEM), each for one combination of selected source language, target language and category. For example, the task *EFB* uses *English Books* reviews as the source language data and uses *French Books* reviews as the target language data.

Table 1: Average classification accuracies (%) and standard deviations (%) over 10 runs for the 18 cross language sentiment classification tasks.

| TASK | TBOW | CL-LSI | CL-KCCA | CL-OPCA | TSL |
|------|------|--------|---------|---------|-----|
| EFB | 67.31±0.96 | 79.56±0.21 | 77.56±0.14 | 76.55±0.31 | **81.92±0.20** |
| FEB | 66.82±0.43 | 76.66±0.34 | 73.45±0.13 | 74.43±0.53 | **79.51±0.21** |
| EFD | 67.80±0.94 | 77.82±0.66 | 78.19±0.09 | 70.54±0.41 | **81.97±0.33** |
| FED | 66.15±0.65 | 76.61±0.25 | 74.93±0.07 | 72.49±0.47 | **78.09±0.32** |
| EFM | 67.84±0.43 | 75.39±0.40 | 78.24±0.12 | 73.69±0.49 | **79.30±0.30** |
| FEM | 66.08±0.52 | 76.33±0.27 | 73.38±0.12 | 73.46±0.50 | **78.53±0.46** |
| EGB | 67.23±0.68 | 77.59±0.21 | 79.14±0.12 | 74.72±0.54 | **79.22±0.31** |
| GEB | 67.16±0.55 | 77.64±0.19 | 74.15±0.09 | 74.78±0.39 | **78.65±0.23** |
| EGD | 66.79±0.80 | 79.22±0.22 | 76.73±0.10 | 74.59±0.66 | **81.34±0.24** |
| GED | 66.27±0.69 | 77.78±0.26 | 74.26±0.08 | 74.83±0.45 | **79.34±0.23** |
| EGM | 67.65±0.45 | 73.81±0.49 | 79.18±0.05 | 74.45±0.59 | **79.39±0.39** |
| GEM | 66.74±0.55 | 77.28±0.51 | 72.31±0.08 | 74.15±0.42 | **79.02±0.34** |
| EJB | 63.15±0.69 | **72.68±0.35** | 69.46±0.11 | 71.41±0.48 | 72.57±0.52 |
| JEB | 66.85±0.68 | 74.63±0.42 | 67.99±0.18 | 73.41±0.41 | **77.17±0.36** |
| EJD | 65.47±0.50 | 72.55±0.28 | 74.79±0.11 | 71.84±0.41 | **76.60±0.49** |
| JED | 66.42±0.55 | 75.18±0.27 | 72.44±0.16 | 75.42±0.52 | **79.01±0.50** |
| EJM | 67.62±0.75 | 73.44±0.50 | 73.54±0.11 | 74.96±0.86 | **76.21±0.40** |
| JEM | 66.51±0.51 | 72.38±0.50 | 70.00±0.18 | 72.64±0.66 | **77.15±0.58** |

**Approaches** We compared the proposed two-step learning (*TSL*) method with the following four methods: TBOW, CL-LSI, CL-OPCA and CL-KCCA. The Target Bag-Of-Word (*TBOW*) baseline method trains a supervised monolingual classifier in the original bag-of-word feature space with the labeled training data from the target language domain. The Cross-Lingual Latent Semantic Indexing (*CL-LSI*) method [10] and the Cross-Lingual Oriented Principal Component Analysis (*CL-OPCA*) method [15] first learn cross-lingual representations with all data from both language domains by performing LSI or OPCA and then train a monolingual classifier with labeled data from both language domains in the induced low-dimensional feature space. The Cross-Lingual Kernel Canonical Component Analysis (*CL-KCCA*) method [20] first induces two language projections by using unlabeled parallel data and then trains a monolingual classifier on labeled data from both language domains in the projected low-dimensional space. For all experiments, we used linear support vector machine (SVM) as the monolingual classification model. For implementation, we used the libsvm package [5] with default parameter setting.

## 5.2 Classification Accuracy

For each of the 18 cross language sentiment classification tasks, we used all documents from the two languages and the additional 2000 unlabeled parallel documents for representation learning. Then we used all documents in the auxiliary source language and randomly chose 100 documents from the target language as labeled data for classification model training, and used the remaining data in the target language as test data. For the proposed method, TSL, we set $\mu = 10^{-6}$ and $\tau = 1$, chose $\gamma$ value from $\{0.01, 0.1, 1, 10\}$, chose $\rho$ value from $\{10^{-5}, 10^{-4}, 10^{-3}, 10^{-2}, 10^{-1}, 1\}$, and chose the dimension $k$ value from $\{20, 50, 100, 200, 500\}$. We used the first task EFB to perform model parameter selection by running the algorithm 3 times based on random selections of 100 labeled target training data. This gave us the following parameter setting: $\gamma = 0.1, \rho = 10^{-4}, k = 50$. We used the same procedure to select the dimensionality of the learned semantic representations for the other three approaches, CL-LSI, CL-OPCA and CL-KCCA, which produced $k = 50$ for CL-LSI and CL-OPCA, and $k = 100$ for CL-KCCA. We then used the selected model parameters for all the 18 tasks and ran each experiment for 10 times based on random selections of 100 labeled target documents. The average classification accuracies and standard deviations are reported in Table 1.

We can see that the proposed two-step learning method, TSL, outperforms all other four comparison methods in general. The target baseline TBOW performs poorly on all the 18 tasks, which implies that 100 labeled target training documents are far from enough to obtain a robust sentiment classifier

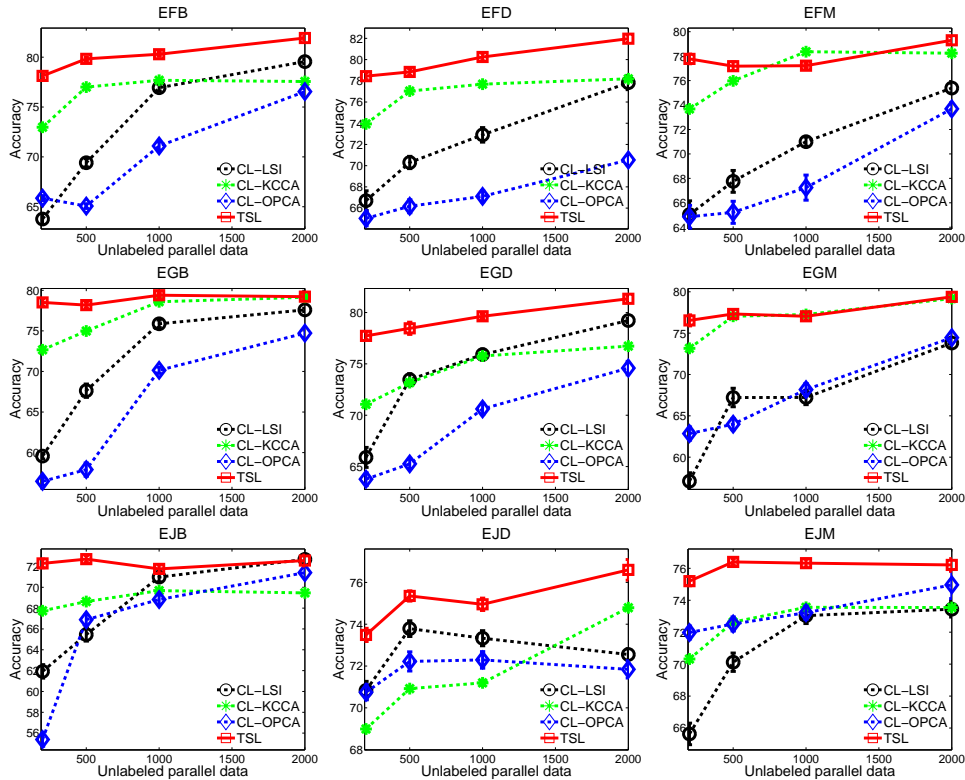

Figure 1: Average test classification accuracies (%) and standard deviations (%) over 10 runs with different numbers of unlabeled parallel documents for adapting a classification system from English to French, German and Japanese.

in the target language domain. All the other three cross-lingual representation learning methods, CL-LSI, CL-KCCA and CL-OPCA, consistently outperform this baseline method across all the 18 tasks, which demonstrates that the labeled training data from the source language domain is useful for classifying the target language data under a unified data representation. Nevertheless, the improvements achieved by these three methods over the baseline are much smaller than the proposed TSL method. Across all the 18 tasks, TSL increases the average test accuracy over the baseline TBOW method by at least 8.59 (%) on the EJM task and up to 14.61 (%) on the EFB task. Moreover, TSL also outperforms both CL-KCCA and CL-OPCA across all the 18 tasks, outperforms CL-LSI on 17 out of the 18 tasks and achieves comparable performance with CL-LSI on the remaining one task (EJB). All these results demonstrate the efficacy and robustness of the proposed two-step representation learning method for cross language text classification.

## 5.3 Impact of the Size of Unlabeled Parallel Data

All the four cross-lingual adaptation learning methods, CL-LSI, CL-KCCA, CL-OPCA and TSL, exploit unlabeled parallel reviews for learning cross-lingual representations. Next we investigated the performance of these methods with respect to different numbers of unlabeled parallel reviews. We tested a set of different numbers, $n_p \in \{200, 500, 1000, 2000\}$. For each number $n_p$ in the set, we randomly chose $n_p$ parallel documents from all the 2000 unlabeled parallel reviews to conduct experiments using the same setting from the previous experiments. Each experiment was repeated 10 times based on random selections of labeled target training data. The average test classification accuracies and standard deviations are plotted in Figure 1 and Figure 2. Figure 1 presents the results for the 9 cross-lingual classification tasks that adapt classification systems from English to French, German and Japanese, while Figure 2 presents the results for the other 9 cross-lingual classification tasks that adapt classification systems from French, German and Japanese to English.

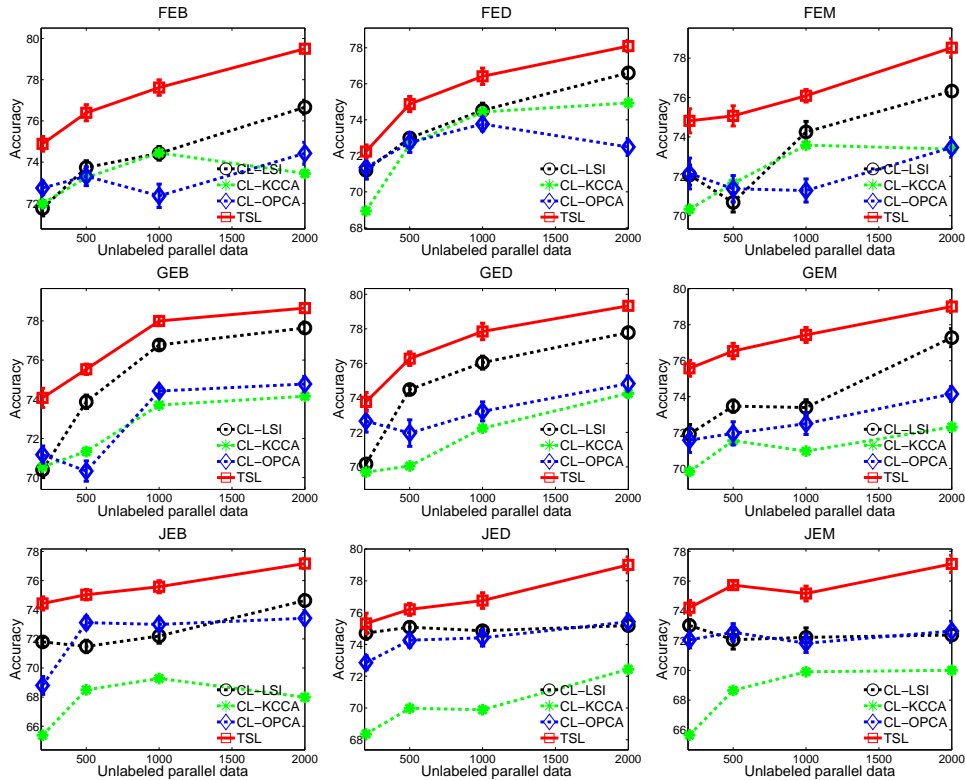

Figure 2: Average test classification accuracies and standard deviations over 10 runs with different numbers of unlabeled parallel documents for adapting a classification system from French, German and Japanese to English.

From these results, we can see that the performance of all four methods in general improves with the increase of the unlabeled parallel data. The proposed method, TSL, nevertheless outperforms the other three cross-lingual adaptation learning methods across the range of different $n_p$ values for 16 out of the 18 cross language sentiment classification tasks. For the remaining two tasks, EFM and EGM, it has similar performance with the CL-KCCA method while significantly outperforming the other two methods. Moreover, for the 9 tasks that make adaptation from English to the other three languages, the TSL method achieves great performance with only 200 unlabeled parallel documents, while the performance of the other three methods decreases significantly with the decrease of the number of unlabeled parallel documents. These results demonstrate the robustness and efficacy of the proposed method, comparing to other methods.

## 6  Conclusion

In this paper, we developed a novel two-step method to learn cross-lingual semantic data representations for cross language text classification by exploiting unlabeled parallel bilingual documents. We first formulated a matrix completion problem to infer unobserved feature values of the concatenated document-term matrix in the space of unified vocabulary set from the source and target languages. Then we performed latent semantic indexing over the completed low-rank document-term matrix to produce a low-dimensional cross-lingual representation of the documents. Monolingual classifiers were then used to conduct cross language text classification based on the learned document representation. To investigate the effectiveness of the proposed learning method, we conducted extensive experiments with tasks of cross language sentiment classification on Amazon product reviews. Our experimental results demonstrated that the proposed two-step learning method significantly outperforms the other four comparison methods. Moreover, the proposed approach needs much less parallel documents to produce a good cross language text classification system.

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
