[Reviews · NeurIPS 2013]

Submitted by Assigned_Reviewer_7

The paper proposes a matrix completion approach to the cross domain classification task, which is capable of exploiting labeled data in an auxiliary domain and also unlabeled parellel data. The approach involves two steps. First, it constructs a single incomplete matrix that includes all documents and domains, and then completes this matrix while enforcing low-rank and sparsity conditions using the projected gradient descent algorithm. Second, it reduces the feature dimension of this completed matrix using LSI, and then trains a standard classifier on this new representation. The paper also presents a convergence guarantee on the projected gradient descent algorithm, and favorable experimental results.

Here are some comments and questions:

- After estimating the completed matrix M*, you reduce the feature dimension from d to k with LSI because M* "lacks sufficient capacity of handling feature sparseness". Can you elaborate on this? Is it basically that you want small dimensions for computational convenience, since M* will be relatively dense even with the sparsity enforcement? Also, in that case, is there a reason you chose LSI over other dimensionality reduction techniques (e.g., PCA)?

- There seem to be two disjoint aspects of exploiting external data in this paper: 1. leveraging the labeled source language data to help learn the labeled target language data, and 2. leveraging the unlabeled parallel data. If this is correct, it would be helpful to clearly distinguish the two and report results separately.

- On a related note, the experiment settings are a bit unclear (section 5.2). So you were using (a) 4k labeled source language documents, (b) 100 labeled target language documents, and (c) 2k unlabeled parallel documents for your training data? And you are reporting the classification results on the remaining 1900 labeled target language documents? Then the only difference in setion 5.3 is that you are varying the number of unlabeled parallel documents from 200 to 2k? In that case, why is the performance of CL-KCCA better than TSL in Figure 1 for EFM and EGM at 2k but it is not in Table 1?

- Can you briefly clarify why a fully observed document-term matrix would be low-rank? It is sparse, but probably no document is a linear combination of other documents in vocabulary, meaning the rank in that case will be min(d, k).

- Can you give an intuition on why TSL is able to perform better in general than other methods, especially CCA? A multiview approach like CCA feels like a natural approach to this cross language representation learning task. In contrast, the matrix completion approach is not as natural. It seems the main driving force is enforcing the low-rank condition on M*. How does this compare with other techniques?

- What is the running time of the algorithm? Does the projected gradient descent algorithm take more, less, or about the same time as other methods like CL-LSI, CL-KCCA, and CL-OPCA?

Here are some suggested changes:

- Fix (ij) to (i,j) two sentences before equation (2).
- Remove "such as" in the sentences before equation (4) and (5).
Summary: The paper proposes a matrix completion approach to the cross domain classification task. Although some technical points in the paper need to be clarified, the method is simple, effective, and in general clearly presented.

Submitted by Assigned_Reviewer_8

Summary: This paper presents a two-step method to learn a cross-lingual topic representation using matrix completion and latent semantic analysis techniques. It uses a projected gradient descent algorithm to optimize the matrix completion problem. Experimental results show that the proposed method outperforms mono-lingual baseline and cross-lingual baselines on sentiment analysis task on parallel Amazon review dataset. Learning curves with different sizes of unlabeled data illustrate that the proposed algorithm learns highly accurate cross-lingual topic representation with relatively small number of parallel data.

Quality: Experiments were thoroughly carried out to support the claim. The propose two-step learning algorithm is carefully compared with two baselines (plain bag-of-words, cross lingual LSA) and two state-of-the-art cross-lingual dimensionality reduction algorithms.

Clarity: This paper is well-organized. All the parameters (including how to select them) were written in the paper; it is easy to reproduce the result. Just a minor comment: I would suggest explaining the motivation of using matrix completion more in Introduction (though it is described in Section 3).

Originality: Although the task of cross-lingual representation learning is not new, the application of matrix completion to this task is novel and it is clear that the proposed algorithm achieves better performance than previous methods in the literature.

Significance: The result of the two-step approach is consistently higher than other baselines. We can learn form this work that matrix completion can be used as a preprocessing for cross-lingual sentiment classification task, and this kind of formulation may be effective in other tasks as well.
Summary: This paper proposes a two-step method to learn a cross-lingual topic representation which perform matrix completion before latent semantic analysis on cross-lingual sentiment classification task. Experimental results show that the proposed method outperforms mono-lingual baseline and other cross-lingual baselines including one of the state-of-the-art methods, and is stable even when trained on different sizes of unlabeled parallel data.

Submitted by Assigned_Reviewer_9

The paper proposes a method for translingual representation learning by first using matrix completion to create multilingual vectors for each row, and then performing CL-LSI on the multilingual vectors.

Quality: While this seems like a good idea, I'm not sure that the results are robust. The positive results could simply be caused by the scaling of LSI being better than OPCA or CCA for a linear SVM.

I would urge the authors to use an ML system that is insensitive to rescalings of the individual feature: for example, boosted decision trees (available in Weka). If the boosted decision trees on top of the two-step system works better than other techniques, then I believe the result.

Significance: If this is really a robust result, it would be neat, because matrix completion is a well-controlled non-linear inference technique.

Novelty: The idea seems new.

Clarity: There were some issues of clarity in the paper. Line 113 talks about "construct[ing] a unified document-term matrix for all documents". I'm not sure whether this means using one term for every surface form (like "Merkel"), or labeling each surface form from each language (like "Merkel_en" and "Merkel_de"). Please clarify. Also, the description of the experimental setup is split across Table 1, so it took me a little while to figure it out.
Summary: Seems like a neat idea. I wish that the authors tried other classifiers than linear SVM -- I just don't know whether the result is robust, or a scaling artifact.
Author Feedback

Author rebuttal: 